# Influence of Recycled Tire Steel Fiber Content on the Mechanical Properties and Fracture Characteristics of Ultra-High-Performance Concrete

**DOI:** 10.3390/ma18143300

**Published:** 2025-07-13

**Authors:** Junyan Yu, Qifan Wu, Dongyan Zhao, Yubo Jiao

**Affiliations:** 1Key Laboratory of Urban Security and Disaster Engineering of Ministry of Education, Beijing University of Technology, Beijing 100124, China; 1245807981@emails.bjut.edu.cn (J.Y.); wuqf573@emails.bjut.edu.cn (Q.W.); b202489097@emails.bjut.edu.cn (D.Z.); 2State Key Laboratory of Bridge Engineering Safety and Resilience, Beijing University of Technology, Beijing 100124, China

**Keywords:** UHPC, recycled tire steel fibers, acoustic emission detection, mechanical properties, fracture properties

## Abstract

Ultra-high-performance concrete (UHPC) reinforced with recycled tire steel fibers (RTSFs) was studied to evaluate its mechanical properties and cracking behavior. Using acoustic emission (AE) monitoring, researchers tested various RTSF replacement rates in compression and flexural tests. Results revealed a clear trend: mechanical properties initially improved then declined with increasing RTSF content, peaking at 25% replacement. AE analysis showed distinct patterns in energy release and crack propagation. Signal timing for energy and ringing count followed a delayed-to-advanced sequence, while b-value and information entropy changes indicated optimal flexural performance at specific replacement rates. RA-AF classification demonstrated that shear failure reached its minimum (25% replacement), with shear cracks increasing at higher ratios. These findings demonstrate RTSFs’ dual benefits: enhancing UHPC performance while promoting sustainability. The 25% replacement ratio emerged as the optimal balance, improving strength while delaying crack formation. This study provides insights into the mechanism by which waste tire steel fibers enhance the performance of UHPC. This research provides valuable insights for developing eco-friendly UHPC formulations using recycled materials, offering both environmental and economic advantages for construction applications.

## 1. Introduction

The mechanical properties and toughness of UHPC are dominated by many factors, among which the incorporation of steel fibers plays a certain role. Characterized by compressive strengths exceeding 150 MPa, coupled with minimal permeability and outstanding abrasion resistance, UHPC delivers superior performance characteristics that make it particularly suitable for demanding engineering applications [1].

Traditional UHPC typically comprises cement, steel fibers, quartz sand, and super-fine fly ash [2]. Researchers have demonstrated that UHPC with steel fibers, as an essential component, can significantly impact the performance of UHPC [3]. Smarzewski et al. investigated the differences between UHPC with added steel fibers and inorganic fibers and UHPC without fibers [4]. The advantageous qualities of steel fibers, such as the ability to restrain crack extension, reduce the extent of stress concentration at crack tips, change crack direction, and slow down crack growth rate, all boost the compressive strength of concrete containing SFs. Some studies show the impact of steel fiber content on the mechanical properties of UHPC and found that the optimal steel fiber content is approximately 2%. At this point, there is a significant improvement in the strength of UHPC [5,6]. Yoo et al. researched the elongation and toughness of UHPC with steel fibers under different loading rates [7]. They performed uniaxial compression tests on samples with and without steel fibers at different loading rates (0.05 mm/min, 0.5 mm/min, and 5 mm/min). The results showed that the addition of steel fibers significantly improved the elongation and toughness of UHPC at all loading rates. Lee et al. studied the impact of steel fibers on the impact resistance of UHPC [8], adding four different volume fractions of steel fibers, 0, 0.5%, 1%, and 1.5%, and found that the addition can significantly improve the impact resistance of UHPC. However, the price of steel fibers is high, and only a tiny volume can increase the cost far beyond that of other composite materials [9]. Therefore, finding a cost-effective steel fiber material becomes a significant concern. During the recycling process of tires, strong, complex, and high-strength steel fibers with good resistance to twisting and yielding can be extracted [10].

Research and surveys have found that China ranks first in the world in terms of waste tire production, with the output increasing year by year. From 3.29 million tons generated in 2004 [11] to 20 million tons today, the volume has grown by nearly 17 million tons in just 20 years. Therefore, the utilization of steel fibers from waste tires exhibits immense potential to reduce environmental pollution and improve economic benefits. Studies have investigated the effects of recycled tire steel fibers on high-strength concrete’s workability and mechanical properties [12,13]. Experiments were carried out with different contents of recycled steel fibers (1%, 2%, and 3%), and it was found they can exhibit similar effects as traditional steel fibers in equivalent bending-tensile strength. Zia A et al. investigated the efficiency of rough steel fibers (RSFs) from recycled tires in improving the durability and strength properties of concrete [14,15,16]. The study found that concrete with RSF addition improved in compressive and tensile strength. Kim et al. studied the application of RSFs in ultra-high fiber-reinforced concrete (UHFRC) [17] and found that using RSFs increased the elongation and post-cracking behavior and improved the tensile performance of HPC at a friction volume greater than 1% RSFs. Zhuo et al. researched the preparation of high-performance concrete using recycled tire fibers [12] and found that adding recycled tire fibers improved the elongation and toughness indices. Serdar et al. investigated the effect of using recycled tire polymer fibers (RTPFs) as a replacement for polypropylene (PP) fibers in concrete to study crack control [18]. The results showed that RTPFs performed similarly to PP in inhibiting internal cracks and did not produce any adverse effects. Zhang et al. explored the influence of RTRFs on the mechanical behavior of UHPC under different loading conditions (including compression, tension, and bending) [19,20]. The results showed that the addition of RTRFs improved the tensile and bending ductility and energy absorption capacity of UHPC but reduced its compressive strength. All of the above studies show the feasibility of replacing steel fibers with RTSFs.

Since UHPC damage occurs internally during loading, it is difficult to observe directly. AE monitoring plays a crucial role in characterizing the damage and cracking of concrete, as the damaged state has significant importance in evaluating the durability and reliability of concrete structures [21,22]. Yue et al. researched the use of AE technology to monitor the tensile behavior of concrete [23,24] and found that AE signals can be used to detect the formation of microcracks and measure the mechanical properties of concrete under tensile loading. Chen et al. used AE technology to analyze the stability of high-performance concrete with different rebar thicknesses and different cement-to-aggregate ratios under four-point bending [25]. The experimental results showed that AE parameters can significantly characterize the evolution process of microcracks and macrocracks in rebar–concrete composite structures. Wu et al. investigated the use of the frequency, amplitude, and duration of AE parameters to assess the damage of concrete under uniaxial compression [26,27]. The results showed that these parameters can be used to identify the durability and onset of damage and estimate the degree of the latter. Yang et al. used AE ring-down counting and energy to study the damage mechanism of polymeric concrete notched samples [28,29] and found that AE parameters can be used to predict the mechanical properties of concrete accurately. The above studies all indicate that AE parameters can reflect the mechanical properties of concrete wells. Currently, research utilizing AE is mainly focused on analyzing the internal damage of concrete structures, while research on the mechanical properties of ultra-high-performance slag concrete (UHPSC) is scarce [30]. Therefore, this paper uses AE to study the influence of old tire steel fibers on the mechanical properties of UHPSC. In previous studies, the relationship between AE parameters and specific mechanical indicators has not been elucidated. Therefore, establishing a connection between AE parameters and concrete mechanics is beneficial for the application of AE in concrete.

As prior research on waste tire steel fibers in UHPC has primarily focused on traditional mechanical properties, employing acoustic emission monitoring to study internal crack progression is essential. This paper investigates the feasibility of replacing steel fibers with RTSFs in ultra-high-performance concrete (UHPC) at optimal aggregate replacement rates, steel fiber content, and lengths. The recycled tire steel fibers are replaced at rates of 25%, 50%, 75%, and 100% to explore the practicability of replacing steel fibers with old tire steel fibers in UHPSC. The mechanical properties of the concrete are analyzed through compressive, bending, and four-point bending tests. Then, the differences in AE parameters and the damage state of UHPSC with different replacement rates of RTSFs are analyzed to establish the relationship between AE parameters and the mechanical properties of UHPSC. This paper explores the feasibility of adding RTSFs to UHPSC, which provides a basis for the development of concrete.

## 2. Materials and Methods

### 2.1. Raw Materials

#### 2.1.1. Basic Raw Materials

Table 1 shows the basic raw materials for preparing UHSPC.

#### 2.1.2. Quartz Sand

The quartz sand is 15–120 mesh. The specific chemical components are shown in Table 2.

#### 2.1.3. Water-Reducing Agent

A polycarboxylic acid efficient water-reducing agent from a design institute in Shandong province was selected, and the specific chemical content is shown in Table 3.

#### 2.1.4. Steel Fibers of Recycled Tires

During the mechanical recycling process of waste tires, the bead wires are separated intact as whole strands, while the steel cords are fragmented into shorter pieces during further processing and then extracted from the rubber using magnets. The bead wires are subsequently processed to produce high-strength steel fibers. The recycled tire steel fibers of a manufacturer in Anyang were selected, and the specific composition content is shown in Table 4. The appearance of the selected recycled tire steel fibers is shown in Figure 1.

### 2.2. Sample Preparation

The laboratory used a small concrete mixing bowl following the GB/T 31387-2015 standard [31,32]. Materials were added separately to avoid uneven mixing, starting with aggregates like cement and slag powder, mixed for 3–4 min. Weighed steel fibers were then added to ensure uniform dispersion. The water-reducing agent was introduced into the water tank, and a slump cone tested the concrete’s fluidity.

After testing, molds were oiled and filled with the mixed material. Three samples were prepared for each group and placed on a vibrating table until the surface was sufficiently slurry. The samples then stood for two days at 15–25 °C. After demolding, they were cured in an environment of 18–22 °C and 95% relative humidity (RH) for 28 days.

### 2.3. Matching of Experimental Raw Materials

According to the study, a steel fiber length of 13 mm, steel fiber yield of 2%, and 10% slag powder replacement rate was used as the best mix to prepare UHPSC. On this basis, steel fibers were replaced at 25%, 50%, 75%, and 100% ratios. RTSF-0 indicates a recycled tire steel fiber replacement ratio of 0%. The materials used for the experiment were combined, as shown in Table 5. The study by Borg et al. [33] showed that the effect of adding recycled fibers on work performance was negligible.

### 2.4. Test Indicators and Methods

#### 2.4.1. Compressive Strength

The concrete specimens were prepared as cubic samples with dimensions of 100 mm × 100 mm × 100 mm, with three samples prepared for each group for a curing period of 28 days. Using a UTM testing machine (Shenzhen Sansi Zongheng Company, Shenzhen, China) with a maximum capacity of 2000 kN, loading was performed at a rate of 0.5–0.8 MPa/s [34]. By calculating the percentage difference between the maximum, minimum, and median values for each group, if the difference did not exceed 15%, the mean value of the three samples was considered representative. If the difference exceeded 15%, the median sample value was taken as the representative value. The experimental setup is shown in Figure 2.

The compressive strength formula is expressed by Equation (1):(1)$$f_{cc}=\frac{F}{A}$$
where *F* is the failure load (N) of the UHPSC cube specimen, and *A* is the pressure area of the UHPSC cube specimen (${mm}^{2}$).

#### 2.4.2. Flexural Strength and Bending Toughness

Per specifications, we prepared three 100 mm × 100 mm × 100 mm rectangular concrete specimens per group for 28-day curing. The MTS testing machine with a capacity of 300 kN was loaded at a rate of 0.08–0.10 MPa/s [35]. After calculating the percentage difference between the maximum, minimum, and median values for each group, if the variation among the maximum, minimum, and median values was ≤15%, we used the mean; otherwise, we used the median. The flexural strength is given by Equation (2):(2)$$f_{f}=\frac{Fl}{bh^{2}}$$
where *F* is the failure load (N) of the UHPSC rectangular specimen, *l* is the support span (mm), *h* is the height of the rectangular mass (mm), and *b* is the rectangular mass width (mm).

The experimental setup is shown in Figure 3.

The load-deflection curve of UHPSC was obtained according to four-point bending tests, recording the relationship between the load and the deviation of the sample from the original position, during which the load-deflection curve [36] was obtained. Analysis of the load-deflection curve is used to deeply understand the performance of concrete materials.

Flexural toughness refers to the ability of a material to deform and resist crack propagation when subjected to bending loads. It reflects the energy absorbed by a material during the bending process and is commonly used to evaluate its crack resistance. Based on the load-deflection curves measured during the four-point bending test, the toughness index for UHPSC can be calculated according to the modified toughness index method outlined in ASTM-C1018 [37]. The flexural toughness is represented by Equation (3):(3)$$\sigma_{b}=\int_{0}^{L}\sigma \cdot \varepsilon \cdot dx$$
where *L* is the specimen length (mm), *σ* is the stress (N) of the specimen under bending load, and *ε* is the strain of the specimen.

The residual strength is the ability of the specimen to resist the external load after a period of use, as indicated by Equation (4):(4)$$\sigma_{r}=\frac{P_{r}\cdot L}{b\cdot d^{2}}$$
where $P_{r}$ is the residual load (kN), *L* is the support span (mm), *b* is the mass width (mm), and *d* is the mass thickness (mm).

The residual strength index is calculated using the maximum load via Equations (5) and (6):(5)$$I_{n}=\frac{A_{n}}{A_{fc}}$$
where $I_{n}$ is the tough index, $A_{n}$ is the area under the load-deflection curve from the initial crack point to the specified deflection, and $A_{fc}$ is the area under the load-deflection curve before the primary crack point.(6)$$R_{5}=I_{5}\times P_{\max}\;R_{10}=I_{10}\times P_{\max}\;R_{20}=I_{20}\times P_{\max}$$
where $P_{\max}$ is the maximum load (kN).

The displacement-load curve of the UHPSC specimen can also be obtained through a four-point bending experiment. During the experiment, the displacement sensor was fixed in the appropriate position of the sample to accurately measure the deformation and analyze the displacement-load curve to further understand the mechanical properties of UHPSC.

Stiffness refers to the ability of a structure to resist deformation when loaded, usually described by the load *P* and the corresponding displacement *δ*, calculated by Equation (7):(7)$$k=\frac{P}{\delta }$$
where *P* is the maximum load of the elastic phase on the displacement-load curve (kN), and *δ* is the corresponding displacement at this time (mm).

The energy absorption value refers to the energy that can be absorbed by a material or structure during the stress process, usually calculated by the area under the curve via Equation (8):(8)$$W=\int_{0}^{L}P\cdot \delta \cdot dx$$
where *P* is the load on the displacement-load curve (kN), and *δ* is the displacement (mm).

#### 2.4.3. Acoustic Emission Detection

Acoustic emission refers to the occurrence of small stress waves within a material when it is subjected to external loads due to factors such as crack propagation, dislocation movement, and grain boundary slip [38]. These stress waves propagate to the surface of the material in the form of ultrasonic waves, which can be captured and analyzed by sensors to assess the material’s fracture state. Acoustic emission technology was employed for real-time monitoring during compressive and flexural strength testing to investigate the damage conditions of UHPSC better under different material actions.

The 12-channel acoustic emission system used threshold triggering (45 dB threshold) with continuous sampling. Sensors were attached to specimen surfaces at fixed intervals using preheated hot-melt glue. The AE sensors were strategically positioned in both the compression and flexural zones of the UHPC specimens to enable real-time monitoring of the failure process.

Figure 4 illustrates the principle of acoustic emission and its related parameters. Complex parameters such as *b*-value, information entropy, and RA-AF values are introduced as follows:

(1)*b*-value: *b*-value was originally developed using the Gutenberg-rich formula [40] for seismology applications. The method has been applied to the acoustic emission of materials and structures and is calculated as shown in Equation (9):

(9)$$\log_{10}N=a*b(\frac{A_{dB}}{20})$$
where $A_{dB}$ is the peak amplitude of the AE hit, and *N* is the cumulative number of AE events with an amplitude greater than $A_{dB}$/20; a is an empirical constant [41].

(2)Information entropy: Information entropy can be used to measure the disorder of information. The concept was originally developed and applied to thermodynamics by Shannon [42], the founder of information theory. The information entropy is low when the system tends to order and higher [43] when the system tends to disorder. In this study, the acoustic emission signals were analyzed by using the information entropy method. According to the entropy calculated by the emission parameters, the damaged state of concrete can be characterized. Adjacent AE events were divided into groups, namely, n = 50. Then, the energy percentage $P_{AE}^{i}$ of different groups was calculated, and the information entropy was calculated as shown in Equation (10):


(10)
$$H(x)=-\sum_{i=1}^{n}P_{AE}^{i}logP_{AE}^{i}$$


(3)RA-AF: RA-AF is a widely used parametric method for classifying fracture modes in AE analysis that relies on average frequency (AF) and rise time to amplitude rate (RA) values. The RA is the rate of rise time to the amplitude of an AE event, expressed in microseconds per volt (ms/V). On the other hand, AF, often called average frequency, represents the count-to-duration rate of an AE event and is measured in kilohertz (kHz) [44]. Numerous AE tests conducted on concrete materials indicate that AF tends to be higher and RA tends to be lower, which is attributed to shorter rise times, brief durations, higher amplitudes, and an increased count of expansion waves produced during tensile cracking [45]. In contrast, shear failure is characterized by longer rise times, durations of shear waves, and smaller amplitudes. When the ringing count is low, the RA value increases while the AF value decreases. Consequently, the RA and AF values can effectively differentiate various crack modes in civil engineering materials and structures [46]. This research used Gaussian mixture models (GMMs) to analyze the probability density associated with the distribution of RA and AF values. By employing the hyperplane generated by a support vector machine (SVM) algorithm, the overlapping clusters of RA and AF can be automatically distinguished [47]. Figure 5 shows the crack classification diagram based on RA and AF values.

## 3. Results and Discussion

### 3.1. Influence of Recycled Tire Steel Fiber Factors on Mechanical Properties

#### 3.1.1. Compressive Strength and Flexural Strength

Figure 6 presents the compressive strength and flexural strength results for UHPSC with varying volumes of recycled tire steel fiber replacements. As the replacement rate increases, the compressive strength decreases due to uneven distribution and fiber agglomeration. Specifically, UHPSC without recycled fibers has a compressive strength of 127 MPa, while full replacement reduces it to 110 MPa. This overall decrease is minor, suggesting that substituting steel fibers with recycled tire steel fibers is feasible for maintaining mechanical performance.

The flexural strength initially increases and then decreases as the replacement rate rises. At a 25% replacement rate, the flexural strength surpasses that of the control group, but it starts to decline beyond this point. This trend occurs because, at lower replacement rates, the fibers are evenly distributed, enhancing bonding and strength. However, at higher rates, fiber agglomeration leads to uneven distribution and reduced strength. Additionally, at optimal replacement levels, the recycled fibers can effectively bridge cracks, improving flexural strength. However, excessive fiber content may weaken the interfacial bond between fibers and the matrix, resulting in decreased reinforcement and lower flexural strength.

#### 3.1.2. Flexural Strength and Bending Toughness

Figure 7 displays the load-deflection curves from the 28-day flexural tests of UHPSC featuring recycled tire steel fibers at equal volume replacement rates. The curves show a similar overall trend, indicating that using recycled fibers is viable. When the replacement ratio of recycled tire steel fibers is 25%, the maximum load-bearing capacity is higher than that of the control group. This is because the rough surface of the recycled tire steel fibers enhances the maximum load-bearing capacity. However, as the replacement ratio continues to increase, the inferior mechanical properties of the recycled tire steel fibers (compared to conventional steel fibers) lead to a decrease in peak load.

At the same time, after adding waste tire steel fibers, the slope of the descending section of the curve increases, indicating that the toughness decreases, and the displacement also decreases, which also reflects a decrease in the bending resistance of UHPSC.

In general, when the replacement rate is 25%, the ultimate load is the largest, and the replacement rate is optimal.

Based on the measured load-deflection curves, the toughness index of UHPSC under different replacement rates of recycled tire steel fibers was calculated using the improved toughness index method according to ASTM-C1018 standards. First, the initial cracking deflection *δ* was determined. From *δ*, the values of 3*δ*, 5.5*δ*, and 10.5*δ* were calculated, allowing for the determination of the flexural toughness indices *I*_5_, *I*_10_, and *I*_20_. The calculations for each index are shown in Table 6.

The results in the table show that when waste tire steel fibers are added, the initial crack load gradually decreases, showing a decrease in strength. When the replacement rate is 25%, the data in the table are not much different from the control group, reflecting that the difference in mechanical properties is not large.

Examining the bending toughness indices (*I*_5_, *I*_10_, *I*_20_) and initial crack load reveals that the replacement rate of tire steel fibers significantly affects the bending toughness of UHPSC beams. At a replacement ratio of 25%, the flexural toughness index shows negligible difference compared to the control group, indicating no degradation in crack resistance. However, when the replacement ratio exceeds 50%, although the toughness index demonstrates some improvement, the marked disparity between the two groups reveals weakened crack propagation resistance.

The residual strength index results in Table 7 show that with the addition of waste tire steel fibers, $R_{5}$ gradually decreased, reflecting a decrease in the ability of samples to maintain basic mechanical properties after bearing load. Meanwhile, when the replacement rate exceeded 25%, the gap between the residual strength indices became larger, indicating a decrease in overall stability.

In conclusion, the selection of the 25% substitution rate has the least influence on the bending performance of UHPSC, and specimens can still maintain their original mechanical properties and overall stability under this substitution rate.

#### 3.1.3. Displacement-Load Curve

Figure 8 shows the displacement-load curves from the four-point bending tests on UHPSC with different volume proportions of recycled tire steel fibers replacing traditional steel fibers. The curves exhibit similar trends during the initial, elastic, and cracking stages, suggesting that recycled tire fibers and steel fibers have comparable characteristics.

At a 25% replacement rate, there is a slight increase in peak load. However, as the replacement rate further increases, the peak strength gradually declines, indicating that the flexural strength of UHPSC follows an initial enhancement followed by deterioration trend with increasing fiber replacement rates. Moreover, as the replacement rate rises, less displacement is needed to reach the peak load, meaning maximum load capacity is achieved with smaller deformations. UHPSC samples with replacement rates below 50% show a slower decline after reaching peak load, while those at 75% to 100% decline rapidly, indicating reduced toughness.

Additionally, the area under the curves indicates that UHPSC with less than 50% replacement has good energy absorption and toughness, whereas rates above 50% exhibit poor energy absorption and decreased toughness.

The stiffness and energy absorption values of UHPSC specimens at different substitution rates from the displacement-load curves are shown as Figure 9.

The addition of RTSFs improved the stiffness of UHPSC specimens, enhancing their resistance to deformation. This improvement is due to the fibers’ ability to enhance the microstructure of UHPC, reduce brittleness, and increase plastic deformation capacity. The fibers also help control crack occurrence and propagation through a pinning effect, which contributes to maintaining high stiffness. The optimal replacement rate for achieving maximum stiffness is 25%, balancing economic and mechanical performance.

However, energy absorption values decrease as the amount of RTSFs increases, indicating a decline in bending toughness. This reduction may result from uneven fiber dispersion within the composite, leading to inconsistencies in internal mechanical properties. Poor interfacial bonding can impede effective energy transfer during deformation and failure, thus lowering energy absorption. While the fibers can enhance initial strength, their toughness may be inadequate for effective energy absorption under high-stress conditions, and high fiber replacement rates can lead to increased brittleness and reduced energy absorption. Therefore, a 25% replacement rate is recommended as the best option.

### 3.2. Analysis of the Influence of Recycled Tire Steel Fiber Factors on the Damage Acoustic Emission Characteristics of UHPSC

AE technology collects the dynamic energy waves generated by a material to assess its status when it is damaged and predict its rupture degree in advance. The parameters used for the evaluation are called the acoustic emission parameters. The destruction process of UHPSC was analyzed using AE parameters to obtain the destruction characteristics of UHPSC under different replacement rates of recycled tire steel fibers.

#### 3.2.1. Compression Characteristics

Figure 10 presents the analysis of acoustic emission energy during compressive tests of UHPSC with varying replacement rates of recycled tire steel fibers. As shown, compared to the control group, the cumulative acoustic emission energy curve of the specimens with recycled tire steel fibers was relatively smooth, indicating no significant internal cracking. As the replacement rate increased to 50%, the cumulative energy rose, suggesting an increase in internal microcracks. When the replacement rate reached 75%, apart from a sudden signal spike at the maximum load, the remaining acoustic emission signals were very low, indicating a shift in the internal cracking mode to brittle failure. At a 100% replacement rate, a similar sudden signal occurred at the maximum load, with minimal acoustic emission signals at other times.

Figure 11 illustrates the correlation of acoustic emission ringing counts during compressive tests of UHPSC with varying replacement rates of recycled tire steel fibers.

Compared to the control group, when the replacement rate of recycled tire steel fibers was 25%, the acoustic emission ring-down count signals decreased, indicating smaller internal cracks and slower crack propagation. This is because the addition of recycled tire steel fibers enhanced the bonding performance of the concrete.

As the replacement rate increased, the acoustic emission signals intensified, suggesting the formation of numerous internal cracks. When the replacement rate reached 100%, the sustained and dense signals indicated rapid internal damage in the specimen, accompanied by a decrease in the maximum load—both reflecting a decline in compressive strength. This occurs because an excessively high replacement rate leads to clustering phenomena inside the specimen, disrupting its original structure and resulting in reduced compressive strength.

Figure 12 analyzes the acoustic emission *b*-value during the compression tests of UHPSC with different replacement rates of recycled tire steel fibers. After incorporating recycled tire steel fibers, the *b*-value remained relatively stable in the early stages compared to the control group, indicating fewer internal cracking events and slower crack development. However, when the replacement rate exceeded 50%, the fluctuations in the *b*-value became more pronounced near the peak load, reflecting a decline in the specimen’s load-bearing capacity.

At a 100% replacement rate, the *b*-value exhibited a decreasing trend from the beginning, suggesting that the specimen’s compressive resistance had already failed. This is attributed to the excessive incorporation of recycled tire steel fibers, which may lead to uneven fiber distribution within the concrete, causing localized fiber clustering. These clusters create stress concentrations, weakening the overall structural integrity and reducing compressive strength.

Combined with the mechanical curve, the specimen exhibits the best compressive performance at a replacement rate of 25%.

#### 3.2.2. Bending Characteristics

Figure 13 compares the AE energy values during the four-point bending tests of UHPC with varying replacement ratios of recycled tire steel fibers. Compared to the control group, at a 25% replacement ratio, the specimens exhibited reduced AE energy during the plastic strengthening phase, accompanied by a slower rise in the cumulative energy curve. This indicates that the incorporation of RTSFs enhances the ductility of UHPC, attributable to the bridging effect of steel fibers within the concrete matrix.

However, when the replacement ratio reached 50%, a sudden jump in the cumulative energy curve occurred toward the end of the strengthening phase, similar to the control group. This phenomenon reflects the formation of extensive internal cracking, signifying an onset of flexural capacity degradation.

Beyond a 50% replacement ratio, the AE energy values remained consistently low, and the cumulative energy curve flattened significantly. This behavior suggests that excessive RTSF content leads to fiber clustering and uneven distribution, increasing the energy required for fiber pull-out. Furthermore, the mechanical response curve demonstrates a rapid strengthening phase followed by abrupt failure, indicating a transition toward brittle fracture behavior.

Figure 14 presents the AE ring count behavior of UHPC specimens under four-point bending tests with varying replacement ratios of recycled tire steel fibers.

For the control group, the ringing count remained stable during Stage I, indicating a dense and homogeneous internal structure. Upon entering Stage II, the cumulative ringing count curve began to rise, reflecting the initiation and propagation of microcracks. In Stage III, the curve stabilized again, suggesting a slowdown in crack development.

At a 25% RTSF replacement ratio, the cumulative ringing count curve exhibited a slower ascent during Stage II, demonstrating delayed crack propagation. This behavior can be attributed to the effective crack-bridging mechanism of the steel fibers, which enhances the flexural performance of UHPC.

However, when the replacement ratio increased to 50%, the ringing count curve rose more rapidly in Stage II, indicating accelerated crack growth. This decline in performance is likely due to fiber clustering and increased porosity caused by excessive RTSF content, which promotes crack formation.

At replacement ratios exceeding 50%, the AE behavior deviated significantly from previous trends. The ringing count remained nearly constant before Stage II, and no pronounced upward trend was observed even during Stage II, suggesting the absence of a plastic strengthening phase and a transition to brittle failure.

Therefore, a 25% replacement rate is considered optimal for improving the flexural strength of UHPSC specimens, as higher rates may lead to increased damage and accelerated crack formation.

Figure 15 shows the acoustic emission *b*-value correlation diagram for the four-point bending tests of UHPSC with different replacement rates of recycled tire steel fibers. From the graph, it can be observed that when no recycled tire steel fibers were added, the fluctuation of the *b*-value was not significant, indicating a relatively stable internal structure. At this time, the crack is dominated by the matrix defect.

At replacement rates of 25% and 50%, the increase in the b-value during the initial stage occurs because the fibers inhibit crack propagation through localized debonding or pull-out, which is a characteristic manifestation of fiber toughening. At this time, the cracks are caused by the superposition of fiber–bulk interface defects. A comparative analysis of the *b*-value evolution reveals significant differences in fracture behavior across replacement ratios. When the replacement ratio increases from 25% to 50%, the *b*-value decreases more rapidly during Stage II, indicating accelerated macrocrack propagation and consequent degradation in flexural performance.

At replacement ratios exceeding 50%, the b-value profile maintains relative stability prior to Stage II, then transitions directly to the failure phase without exhibiting the characteristic strengthening stage—a clear signature of brittle fracture. This behavioral shift is attributed to interfacial deterioration between the concrete matrix and steel fibers caused by excessive fiber content, which leads to fiber clustering, stress concentration, and ultimately loss of composite integrity.

Figure 16 illustrates the correlation of acoustic emission information entropy during the four-point bending tests of UHPSC at different replacement rates of recycled tire steel fibers. When no recycled tire steel fibers were added, the information entropy exhibited a declining trend in Stage I, indicating that the internal elastic deformation mechanism of the specimen was relatively singular. In Stage II, the entropy showed intense fluctuations, reflecting rapid crack propagation and increased instability within the material.

When the replacement ratio was below 50%, the entropy displayed an upward trend in Stage I, demonstrating that the incorporation of recycled tire steel fibers enhanced the complexity of the material’s microstructure. During the initial loading phase, interface friction and localized debonding between the fibers and matrix contributed to the increase in entropy.

In Stage II, specimens with a 25% replacement ratio exhibited smaller entropy fluctuations compared to those with 50%, suggesting a more compact internal structure and consequently improved flexural performance.

When the replacement ratio exceeded 50%, the entropy showed a continuous decline. This was attributed to excessive fiber content causing the material structure to become more homogenized, with fiber fracture emerging as the dominant damage mechanism, thereby reducing entropy. Correlation with mechanical behavior: The observed entropy trends align with the stress–strain curves, where specimens exceeding 50% fiber replacement exhibited brittle failure characteristics, further validating the structural degradation due to fiber overdosage.

Figure 17 illustrates the RA-AF relationship of acoustic emission during the four-point bending tests of UHPSC at different replacement rates of recycled tire steel fibers.

When no waste tire steel fibers were added, the proportion of shear cracks was 67.40%, indicating that shear failure was dominant at this stage. When the replacement ratio of recycled tire steel fibers (RTSFs) was 25%, the proportion of shear cracks decreased to 60.00%, indicating a reduction in internal shear failure. This improvement can be attributed to the enhanced ductility of specimens due to the incorporation of RTSFs. However, as the replacement ratio further increased, the proportion of shear cracks gradually rose. This phenomenon occurred because contaminants on the surface of the recycled fibers might have formed a porous interfacial transition zone (ITZ), which becomes a preferential failure path under shear stress.

## 4. Conclusions

This paper explores how different replacement rates of recycled tire steel fibers affect the mechanical properties and acoustic emission characteristics of UHPSC. We evaluated the compressive and flexural strengths of UHPSC by measuring compressive strength, flexural strength, and displacement-load curves. Additionally, we analyzed the fracture process of UHPSC using acoustic emission parameters from compressive and four-point bending tests. From this study, we can draw the following conclusions.

(1)At the 25% replacement ratio, UHPC achieves peak flexural strength while maintaining an acceptable compressive strength reduction. This ratio provides an optimal balance of key mechanical properties.(2)As the recycled tire steel fiber replacement rate increases, the bending toughness index also shows a gradual decrease. At 25%, there is no significant difference with the control group. When the replacement rate exceeds 25%, the gap between the indices becomes larger, indicating that the bearing deformation ability decreases.(3)The stiffness of UHPSC specimens increases overall with the addition of recycled tire steel fibers, peaking at a 25% replacement rate, indicating improved deformation resistance. Conversely, energy absorption decreases with higher replacement rates. Thus, considering both stiffness and energy absorption, 25% is determined to be the optimal replacement rate.(4)Acoustic emission analysis revealed that recycled tire steel fibers effectively inhibit crack formation in UHPSC. At 25% replacement, signals increased gradually, indicating controlled crack growth. Excessive replacement (>50%) caused signal anomalies, signaling brittle failure transition.(5)Optimal recycled tire steel fiber addition enhances UHPSC flexural performance. At 25% replacement, delayed AE energy/ringing counts and improved b-value/entropy levels indicate better crack resistance. When the replacement rate exceeded 50%, the acoustic emission signals exhibited drastic fluctuations at peak load while demonstrating minimal signal activity prior to this stage, collectively indicating a transition in the internal failure mode toward brittle fracture.(6)Based on the *b*-value and RA-AF correlation analysis, it can be concluded that adding an appropriate amount of recycled tire steel fibers alters the dominant crack propagation mode. At a 25% replacement ratio, the proportion of shear cracks decreases, while the *b*-value increases during the initial stage. Both observations indicate a transition from shear-dominated failure to a mixed tensile–shear mode, which improves stress redistribution at the crack tip and mitigates the inherent brittleness of the matrix.

This paper investigates the mechanical properties and fracture characteristics of UHPSC at different replacement rates of recycled tire steel fibers using acoustic emission technology. The findings provide guidance for the preparation of ultra-high-strength concrete using new materials. However, the effects of the length and thickness of recycled tire steel fibers still require further investigation. Future research should focus on exploring how fiber length and thickness influence the stability of UHPSC.

## Figures and Tables

**Figure 1 materials-18-03300-f001:**
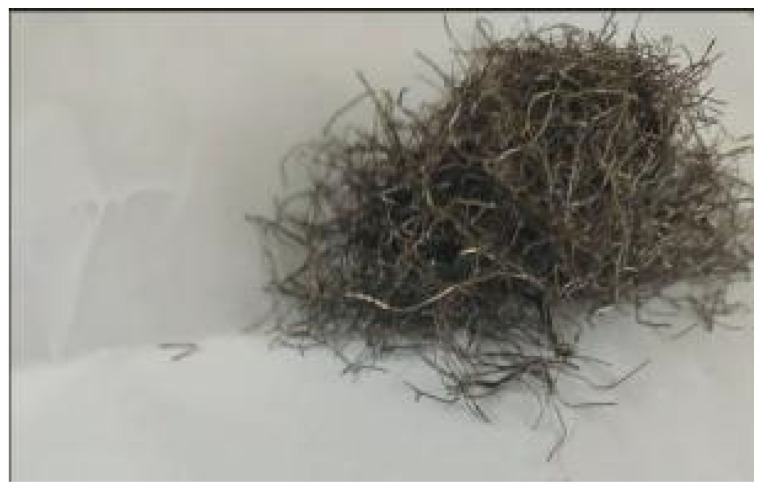
Appearance of recycled tire steel fibers.

**Figure 2 materials-18-03300-f002:**
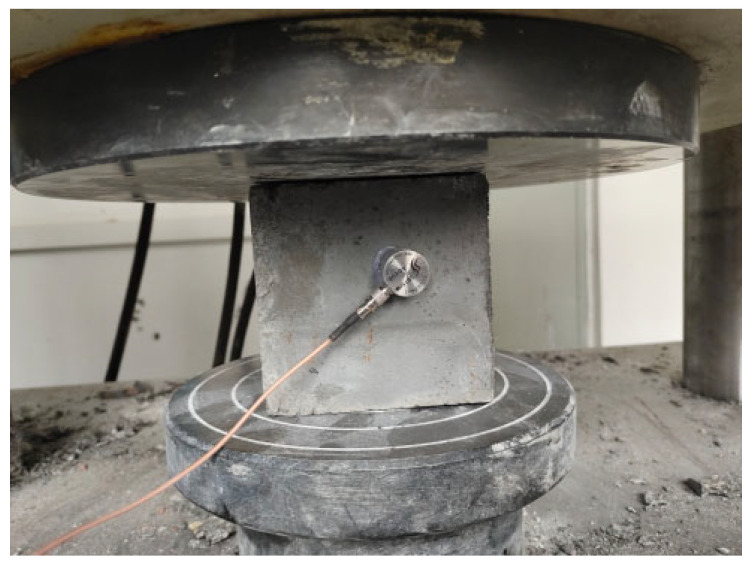
Compressive strength test.

**Figure 3 materials-18-03300-f003:**
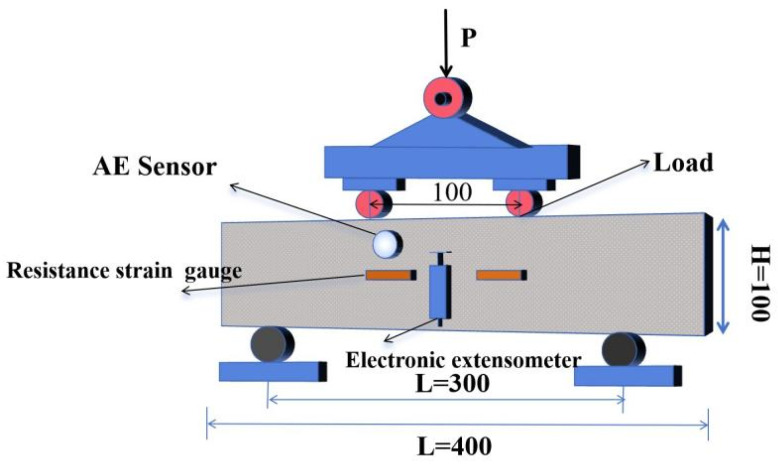
Four-point bending test (unit: mm).

**Figure 4 materials-18-03300-f004:**
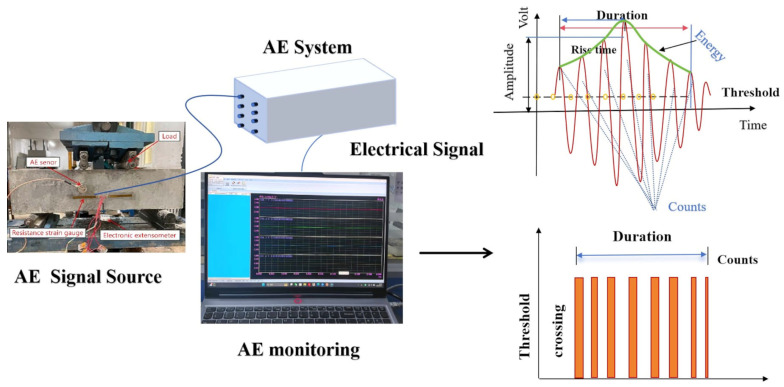
The principle of acoustic emission and its related parameters [39].

**Figure 5 materials-18-03300-f005:**
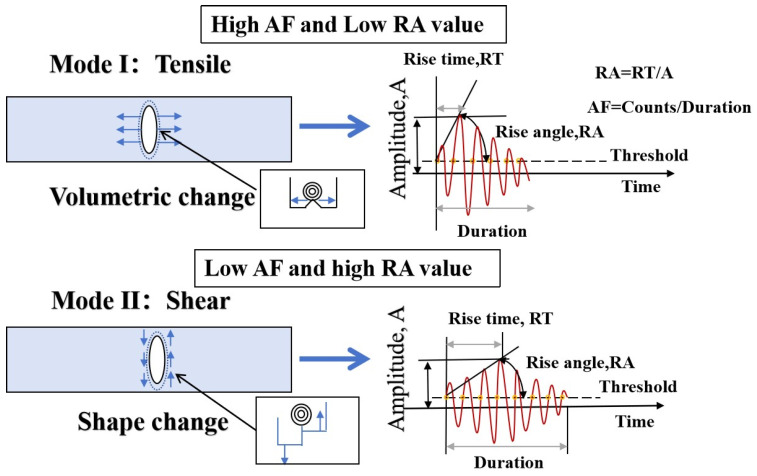
Schematic diagram of crack classification based on RA and AF parameters.

**Figure 6 materials-18-03300-f006:**
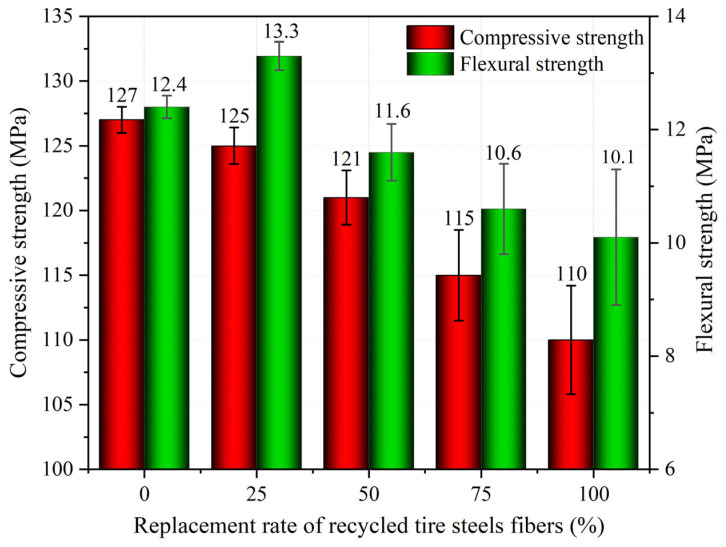
Effect of recycled tire steel fiber replacement rate on compressive strength and flexural resistance.

**Figure 7 materials-18-03300-f007:**
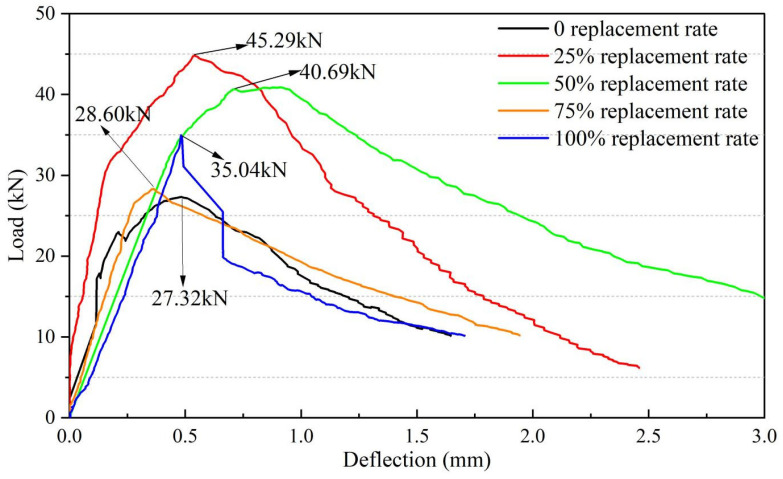
Load-deflection curves of different recycled tire steel fiber replacement rates.

**Figure 8 materials-18-03300-f008:**
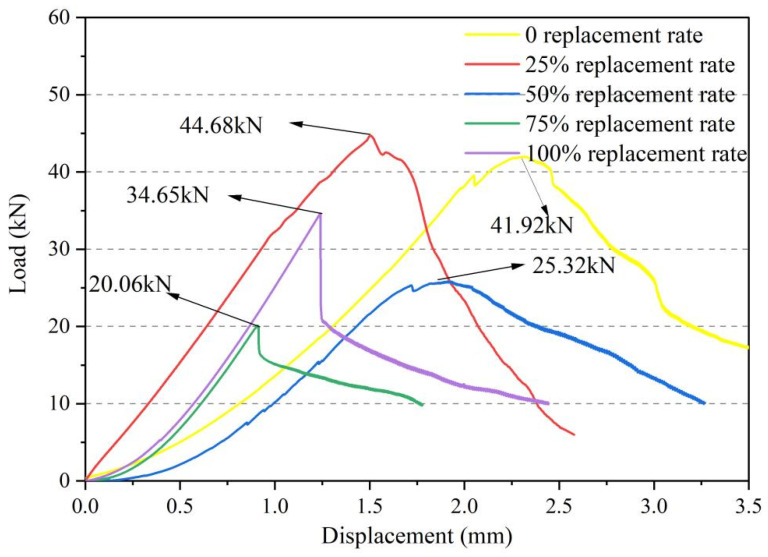
Displacement-load curves of different recycled tire steel fiber replacement rates.

**Figure 9 materials-18-03300-f009:**
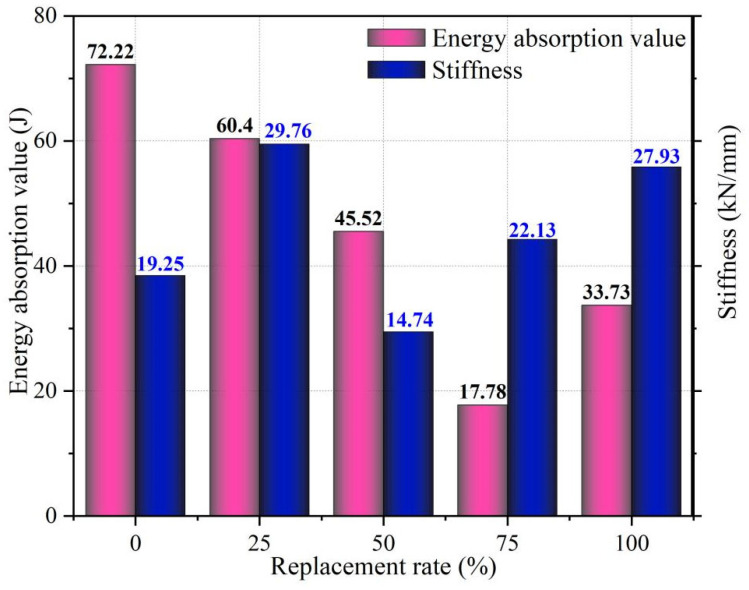
Stiffness and energy absorption values of UHPSC specimens at different substitution rates.

**Figure 10 materials-18-03300-f010:**
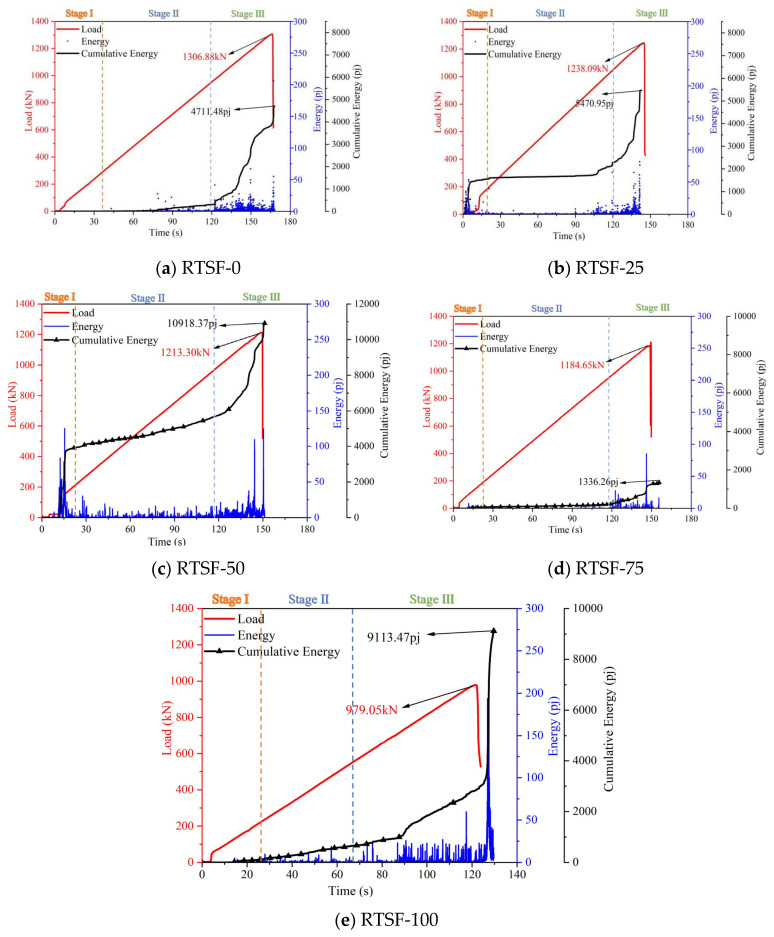
Comparison of acoustic emission energy in compressive tests of UHPSC with different replacement rates of recycled tire steel fibers.

**Figure 11 materials-18-03300-f011:**
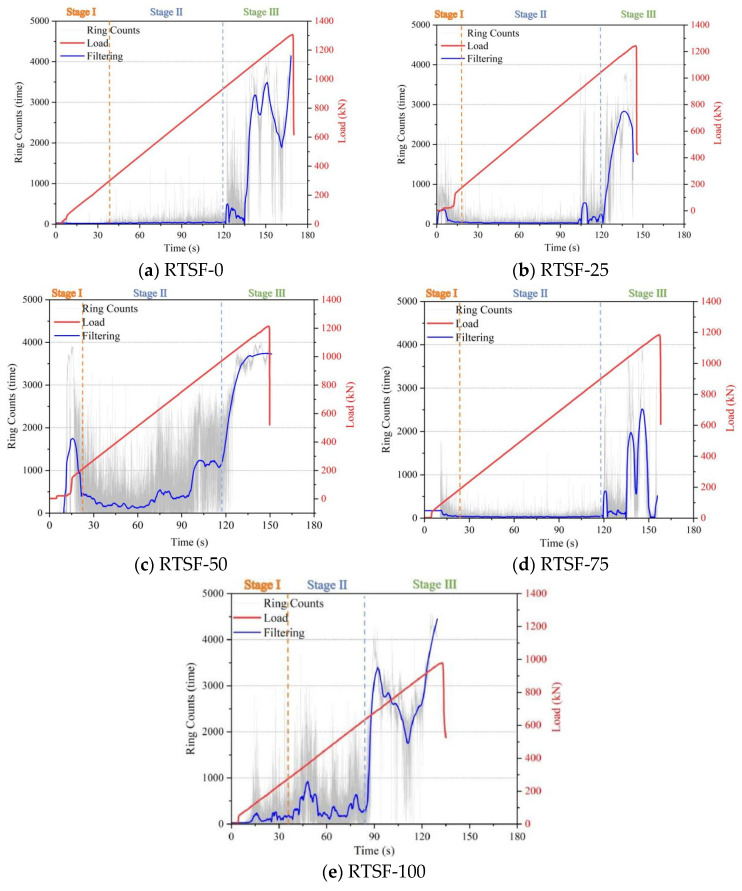
Comparison of acoustic emission ringing counts in compressive tests of UHPSC with different replacement rates of recycled tire steel fibers.

**Figure 12 materials-18-03300-f012:**
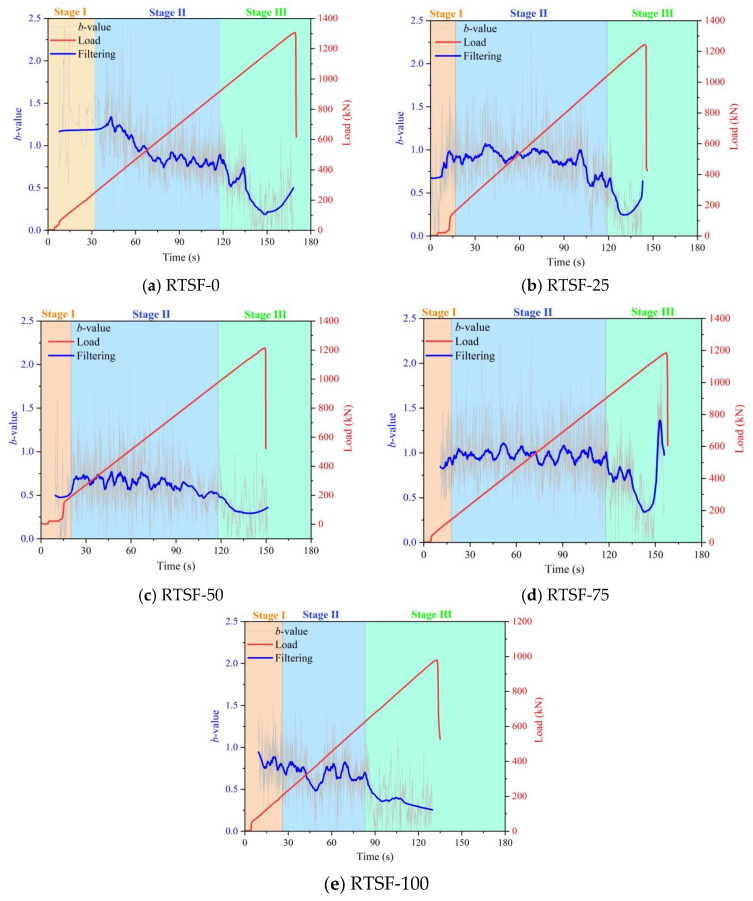
Comparison of acoustic emission *b*-values in compressive tests of UHPSC under different replacement rates of recycled tire steel fibers.

**Figure 13 materials-18-03300-f013:**
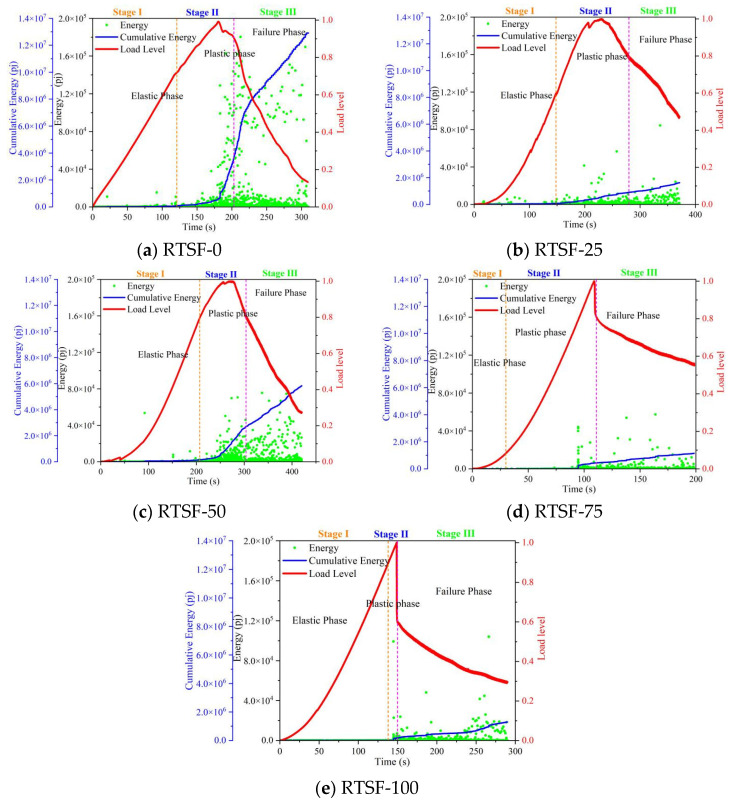
Comparison of acoustic emission energy values during four-point bending tests of UHPSC under different replacement rates of recycled tire steel fibers.

**Figure 14 materials-18-03300-f014:**
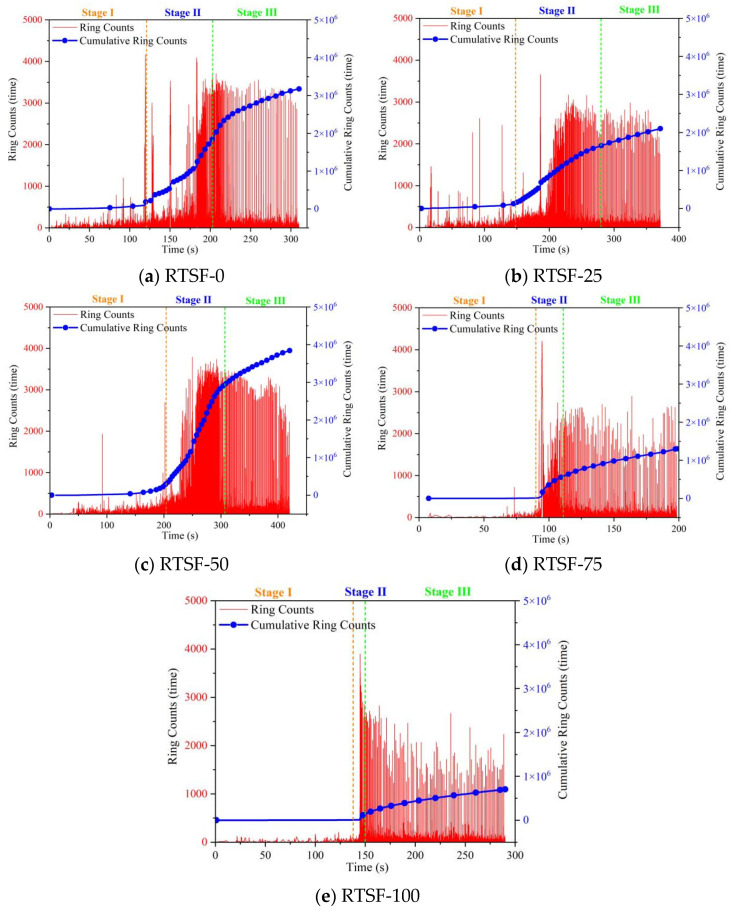
Comparison of acoustic emission ringing counts during four-point bending tests of UHPSC under different replacement rates of recycled tire steel fibers.

**Figure 15 materials-18-03300-f015:**
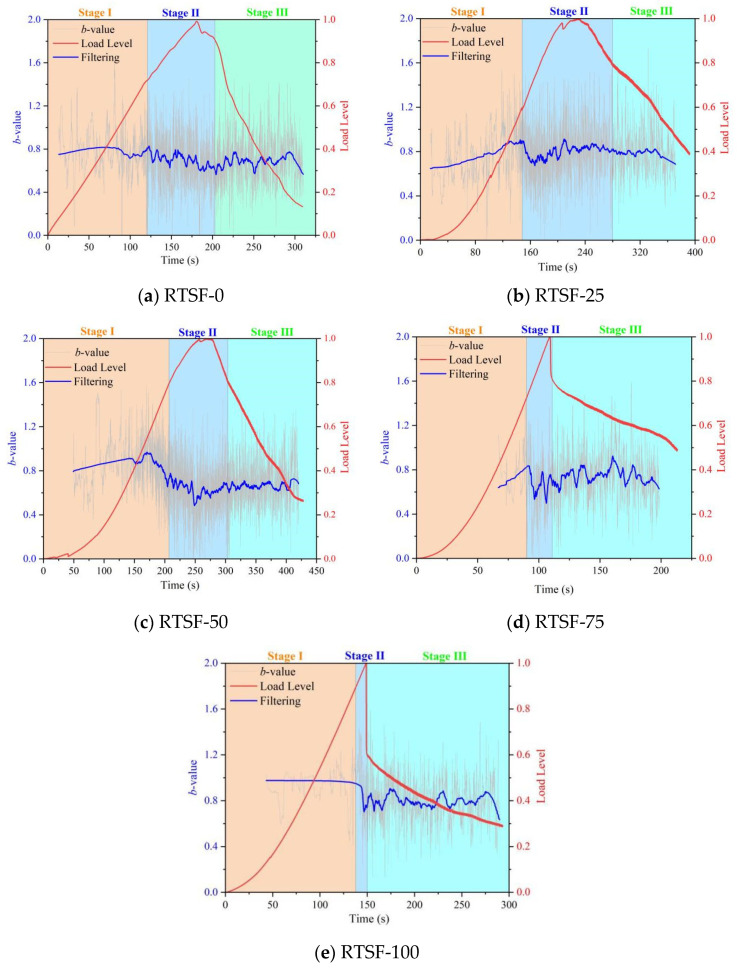
Control diagram of AE *b* values in UHPSC four-point bending tests at different used tire steel fiber replacement rates.

**Figure 16 materials-18-03300-f016:**
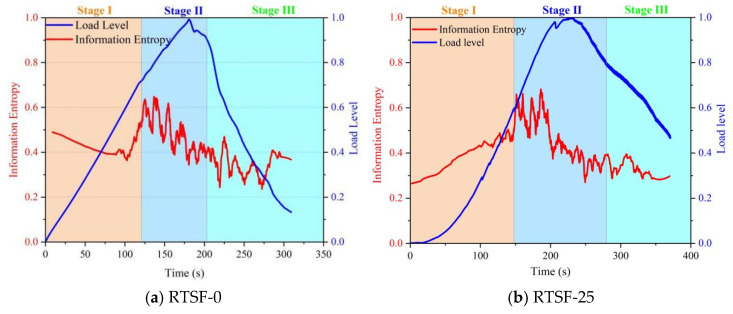
Control diagram of AE information entropy in UHPSC four-point bending tests at different used tire steel fiber replacement rates.

**Figure 17 materials-18-03300-f017:**
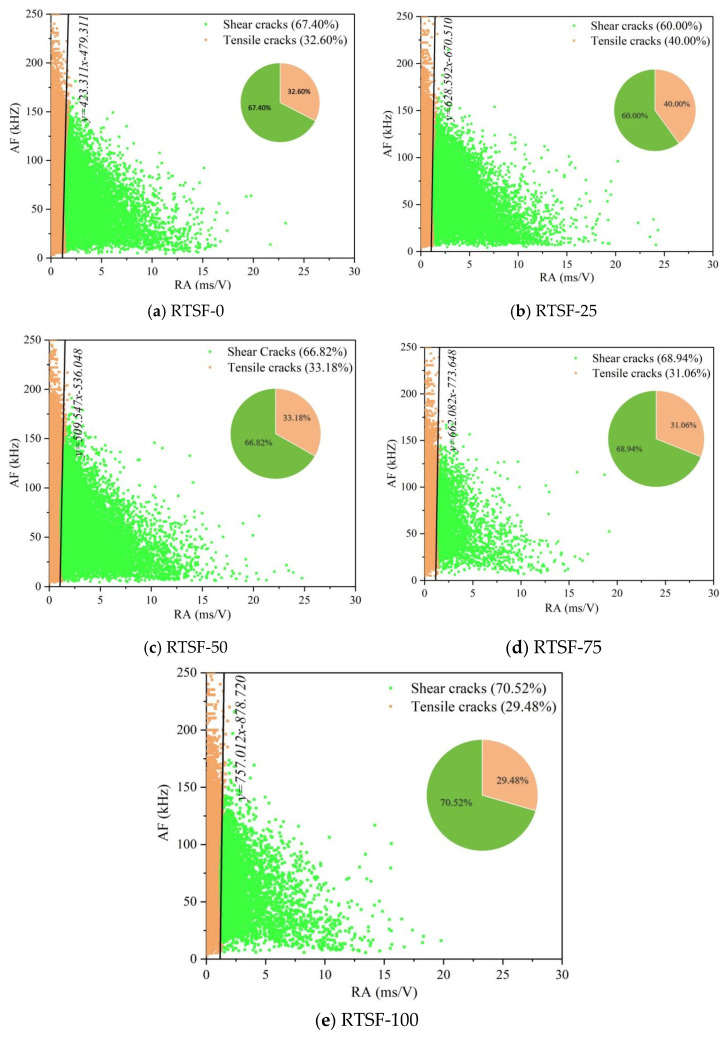
Control diagram of AE RA-AF in UHPSC four-point bending tests at different used tire steel fiber replacement rates.

**Table 1 materials-18-03300-t001:** Basic raw material composition.

Material Categories	Component	CaO	SiO_2_	Al_2_O_3_	SO_3_	Fe_2_O_3_	MgO
Cement	Wt/%	49.70	22.60	9.87	3.84	3.50	2.06
Silica fume	Wt/%	1.86	94.8	0.81	0.45	0.08	0.65
Quartz powder	Wt/%	0.023	99.0	0.22	0.056	0.015	0.011

**Table 2 materials-18-03300-t002:** Composition of quartz sand.

Component	Soluble Rate of Hydrochloric Acid	SiO_2_	Sediment Percentage	Percentage of Damage	Porosity	Rate of Wear
Wt/%	≤3.0	99.3	≤1.0	0.51	43	0.35

**Table 3 materials-18-03300-t003:** Chemical composition content of the water-reducing agent.

Index	NaSO_4_ (%)	PH	Cl (%)	Fineness (%)	Water Reduction Rate (%)	Air Content (%)
numerical value	0.32	5.60	0.02	0.20	26	3

**Table 4 materials-18-03300-t004:** Recycled tire steel fiber composition content.

Component	Length (mm)	Tensile Strength (MPa)	Density (g/cm^3^)	Diameter (mm)
Wt/%	13	2850	7.80	0.17

**Table 5 materials-18-03300-t005:** Mix-rate design of the experimental materials.

Experimental Group Number	RTSF-0	RTSF-25	RTSF-50	RTSF-75	RTSF-100
Cement	765	765	765	765	765
Silica fume	195	195	195	195	195
Quartz sand	935	935	935	935	935
Breeze	85	85	85	85	85
Quartz flour	331.50	331.50	331.50	331.50	331.50
Steel fibers from used tires	0	39.25	78.5	117.75	157
Steel fibers	157	117.75	78.5	39.25	0
Water	167.28	167.28	167.28	167.28	167.28
Water-reducing agent	51	51	51	51	51

**Table 6 materials-18-03300-t006:** The calculation results of bending toughness.

Specimen Number	*F_L_*/kN	*δ*/mm	3*δ*	5.5*δ*	10.5*δ*	*I* _5_	*I* _10_	*I* _20_
RTSF-0	12.011	0.116	0.348	0.638	1.218	7.349	16.816	30.737
RTSF-25	31.898	0.177	0.531	0.974	1.859	4.797	9.904	15.615
RTSF-50	34.896	0.503	1.509	2.767	5.032	5.928	9.728	10.920
RTSF-75	5.0701	0.061	0.183	0.336	0.641	8.211	26.146	65.225
RTSF-100	3.938	0.085	0.255	0.468	0.893	9.096	32.860	77.200

**Table 7 materials-18-03300-t007:** Results of the residual intensity calculation.

Specimen Number	$R_{5}$	$R_{10}$	$R_{20}$
RTSF-0	70.624	161.601	295.381
RTSF-25	88.012	181.711	286.492
RTSF-50	138.976	228.063	256.008
RTSF-75	54.288	172.866	431.240
RTSF-100	70.688	255.364	599.944

## Data Availability

The original contributions presented in this study are included in the article. Further inquiries can be directed to the corresponding author.

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
