# Peer review of "Influence of Recycled Tire Steel Fiber Content on the Mechanical Properties and Fracture Characteristics of Ultra-High-Performance Concrete"

_materials, 2025, doi:10.3390/ma18143300_

Round 1
Reviewer 1 Report
Comments and Suggestions for Authors
Please reformat the manuscript as references/tables/figures are not correct. There are also spelling mistakes. Some of the details are too long to read - please try to be succinct. Some other comments are as follows:
- Line 28 – “The outstanding mechanical properties and toughness of UHPC are largely attribut-27 able to the incorporation of steel fibers” – this line may not necessarily be accurate. The mechanical properties and toughness depends on so many other factors including the matrix.
- Line 38 – Please be specific – “varying degrees” is very vague.
- Line 118 – Please clarify the meaning of “The quartz sand is 15-120 items”
- Please be consistent with terminology. Section 2.1.1. says “silicon ash” whereas table 7 says silica fume. Same comment for quartz flour. Also, specify the property of “breeze”.
- Table 7 – The dosage of water reducer is same for all mixes – does this mean that the use of recycled fibres didn’t affect the workability? How was it ensured?
- Line 177, 320, 338,411 – “Error! Reference source not found.”
- Please shorten the details in the testing section. Most of the details are quite standard. Please try to be succinct and avoid repetitions.
- Figure 7 – Please include error bar (Standard deviation) in the data as the values are quite close.
- Line 295 – How the flexural strength increased at 25% replacement even with uneven distribution (and compressive strength decrease)?
- Line 302- What is “curling strength”?
- 8 – From the figure it appears that black line (0 percent replacement) had the lowest peak load? In fact, the trend is looking just opposite. Please clarify this.
- Please clarify what is the difference between Fig. 8 and Fig. 9 and explain clearly. It would be better to merge both in same section as both are bending test curves.
- The section on AE is too long – it is recommended to present the discussion succinctly. Figure 11 and 12 presents similar information. Same comment for other later figures. Please combine figures which do not need to be added separately. Also correct figure numbers as they seem to be wrong so it is very hard to follow through.
Author Response
Thank you for your valuable suggestions on my manuscript. I found them very helpful and have revised the manuscript accordingly based on your feedback. The revised version with tracked changes has been submitted via Word document.

Reviewer 2 Report
Comments and Suggestions for Authors
It is necessary to highlight the novelty of the research.
Since measurements of recycled tire steel fiber content are reported.
Measured values ​​of similar studies should be reported and the novelty of the paper should be highlighted.
Exhausted tires are recycled in many sectors from acoustics for sound-absorbing panels and for insulating material.
Or vibration-insulating material.
It should be better explained how you make your material. The element of novelty and interest for readers.
Better highlight how you perform the measurements.
Improve the quality of the figures
Author Response

(The authors gave the same response as above.)

Round 2
Reviewer 1 Report
Comments and Suggestions for Authors
The authors have addressed the comments well. Some minor comments.
(Previous comment 8) Error bar provided in Figure 6 does not seem accurate. It looks like all the data had same deviation which is very unlikely. Please recheck this and correct the figure.
